# An exploration into CTEPH medications: Combining natural language processing, embedding learning, *in vitro* models, and real-world evidence for drug repurposing

**Daniel Steiert**[1☯], **Corey Wittig**[1☯], **Priyanka Banerjee**[2], **Robert Preissner**[2], **Robert Szulcek**[1,3]*

**1** Laboratory of in vitro modeling systems of pulmonary and thrombotic diseases, Institute of Physiology, Charité–Universitätsmedizin Berlin, corporate member of Freie Universität Berlin and Humboldt-Universität zu Berlin, Berlin, Germany, **2** Structural Bioinformatics Group, Institute of Physiology, Charité–Universitätsmedizin Berlin, corporate member of Freie Universität Berlin and Humboldt-Universität zu Berlin, Berlin, Germany, **3** Deutsches Herzzentrum der Charité, Department of Cardiac Anesthesiology and Intensive Care Medicine, Berlin, Germany

☯ These authors contributed equally to this work.
* robert.szulcek@charite.de

**Data Availability Statement:** The datasets or analyzed for/or analyzed for this study are available in the Dataverse repository,

## Abstract

### Background

In the modern era, the growth of scientific literature presents a daunting challenge for researchers to keep informed of advancements across multiple disciplines.

### Objective

We apply natural language processing (NLP) and embedding learning concepts to design PubDigest, a tool that combs PubMed literature, aiming to pinpoint potential drugs that could be repurposed.

### Methods

Using NLP, especially term associations through word embeddings, we explored unrecognized relationships between drugs and diseases. To illustrate the utility of PubDigest, we focused on chronic thromboembolic pulmonary hypertension (CTEPH), a rare disease with an overall limited number of scientific publications.

### Results

Our literature analysis identified key clinical features linked to CTEPH by applying term frequency-inverse document frequency (TF-IDF) scoring, a technique measuring a term's significance in a text corpus. This allowed us to map related diseases. One standout was venous thrombosis (VT), which showed strong semantic links with CTEPH. Looking deeper, we discovered potential repurposing candidates for CTEPH through large-scale neural network-based contextualization of literature and predictive modeling on both the CTEPH and

https://doi.org/10.7910/DVN/TWZAGW. Please refer to the "Dataset_description.txt" file and the individual file annotations in the repository for a detailed description of the files and datasets. Our tool, PubDigest, and its documentation can be accessed in the GitHub repository: https://github.com/dansteiert/PubDigest.

**Funding:** We acknowledge financial support from the Open Access Publication Fund of Charité – Universitätsmedizin Berlin and the German Research Foundation (DFG) for covering parts of the publication fees. The funders had no role in study design, data collection and analysis, decision to publish, or preparation of the manuscript.

**Competing interests:** The authors have declared that no competing interests exist.

the VT literature corpora to find novel, yet unrecognized associations between the two diseases. Alongside the anti-thrombotic agent caplacizumab, benzofuran derivatives were an intriguing find. In particular, the benzofuran derivative amiodarone displayed potential anti-thrombotic properties in the literature. Our *in vitro* tests confirmed amiodarone's ability to reduce platelet aggregation significantly by 68% (p = 0.02). However, real-world clinical data indicated that CTEPH patients receiving amiodarone treatment faced a significant 15.9% higher mortality risk (p<0.001).

## Conclusions

While NLP offers an innovative approach to interpreting scientific literature, especially for drug repurposing, it is crucial to combine it with complementary methods like *in vitro* testing and real-world evidence. Our exploration with benzofuran derivatives and CTEPH underscores this point. Thus, blending NLP with hands-on experiments and real-world clinical data can pave the way for faster and safer drug repurposing approaches, especially for rare diseases like CTEPH.

### Author summary

We tackled the challenge of keeping up with the ever-growing scientific literature. We focused on leveraging the power of natural language processing (NLP) to work through extensive literature data, targeting the discovery of new drug applications. Our tool, Pub-Digest, applies advanced NLP techniques and scans vast amounts of research abstracts from PubMed to uncover links between drugs and diseases (Fig 1). Our primary case study presents chronic thromboembolic pulmonary hypertension (CTEPH), a rare but life-threatening condition. Employing PubDigest, a notable discovery was the potential use of caplacizumab or benzofuran derivatives like amiodarone in treating CTEPH, suggested by their anti-thrombotic properties. However, we didn't rely solely on computational analysis. Lab experiments, toxicity prediction, and clinical data were essential to validate these findings. While amiodarone showed promise in laboratory tests, real-world clinical data indicated increased mortality risks in CTEPH patients treated with it. This highlighted the crucial need to complement computational discoveries with practical, real-world evaluations. Our research shows the value of combining computational tools with traditional methods in medical research. This approach accelerates drug discovery and emphasizes a comprehensive view, especially for rare diseases. Through this work, we aim to contribute a novel, efficient pipeline for drug discovery that opens new doors in medical treatment and research.

## Introduction

The modern scientific landscape has witnessed a monumental surge in literature over the past two decades. Specifically, MEDLINE has observed an average addition of approximately 770,000 new citations annually [1]. Notably, the years 2020 to 2022 witnessed an unprecedented spike, with each year recording close to one million new citations, doubling the figures from the 2000s. Such a consistent and at times overwhelming increase, averaging about 2,700

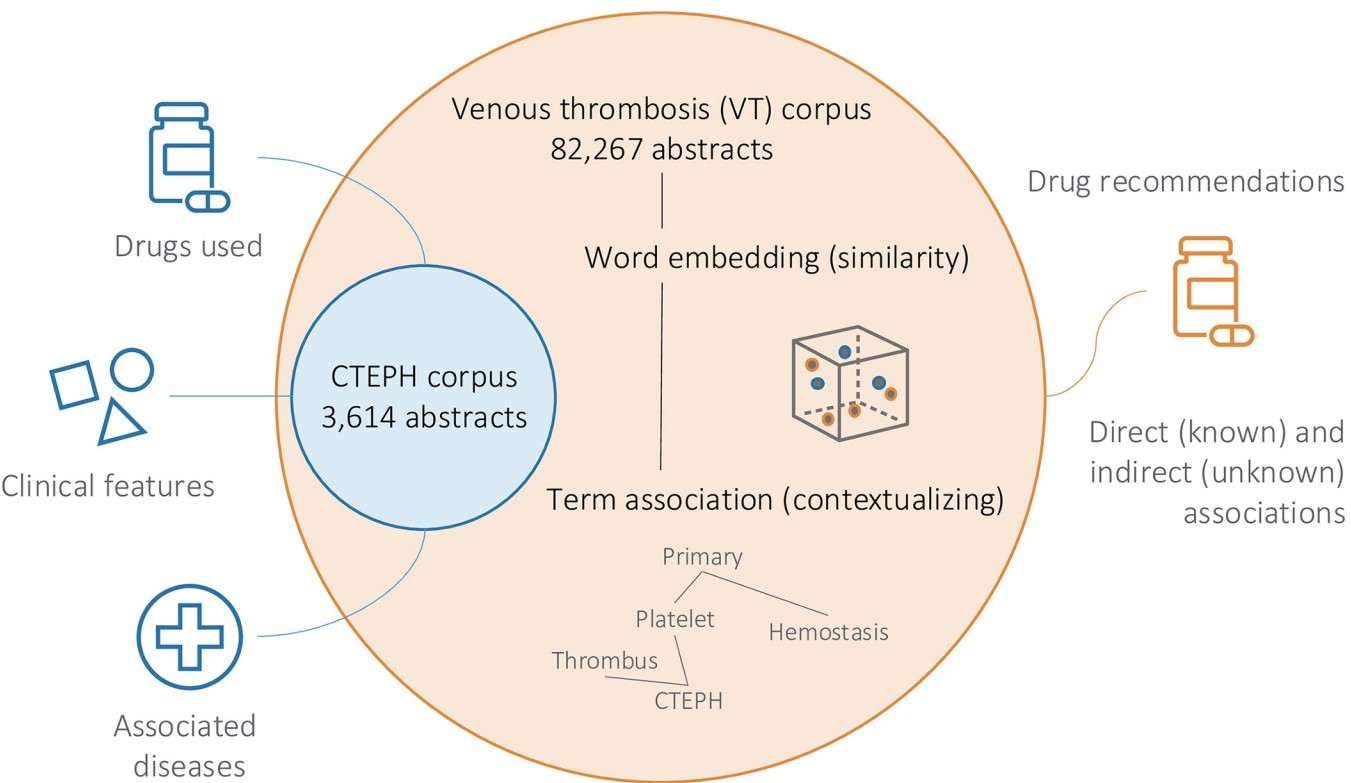

**Fig 1. Identifying drug repurposing candidates for CTEPH.** PubDigest aims to automatically identify drug repurposing candidates from scientific literature. Approximately 3,600 abstracts referencing "chronic thromboembolic pulmonary hypertension" (CTEPH), a rare lung disease, were downloaded from the PubMed database. These abstracts were automatically analyzed to identify drugs used for, clinical features associated with, and diseases semantically linked to CTEPH. The CTEPH literature corpus was integrated with the corpus of venous thrombosis (VT), one of the semantically related conditions, consisting of around 82,000 published abstracts. By applying natural language processing (NLP) and embedding learning concepts, such as word embeddings and term associations, drugs from the VT field were identified that might have potential for repurposing in CTEPH. One of these identified candidates was followed up with *in vitro* and *in silico* experimentations and real-world data evaluation.

new citations daily for 2022, makes extracting critical insights and integrating these into existing research a significant challenge.

This especially holds true for drug repurposing, which is a vital approach for its potential in cost-saving and time efficiency that is often employed to address urgent therapeutic needs, especially in rare diseases with lower research interest and limited access to human samples and/or funding.

Historically, systematic literature reviews have been the primary tool used to encapsulate the essence of a research field, setting impulses for future research. However, this approach is burdened with challenges in the modern era. The process entails a time and labor-intensive manual collection and summarization of hundreds of articles, often suffers from potential publication and interpretation biases, and lacks longevity in utility due to not being systematically updated [2]. Moreover, the extensive duration of the literature collection, interpretation, and writing process, coupled with lengthy publication cycles and the fast appearance of new literature often results in reviews falling behind the current or evolving discourse. As such, reviews rarely uncover new insights or relationships, and their creation necessitates significant expertise in the subject matter.

To overcome these hurdles, computational tools like natural language processing (NLP) represent an exciting approach. Knowledge graphs, for instance, which map entities and their

relationships, have shown promise in revealing connections in fields like biomedical and life sciences [3,4] and are regularly employed to identify drugs for repurposing based on molecular actions, pathways, or drug-disease descriptors [5,6]. These graphs can adapt and grow, with techniques like link prediction and machine learning strategies to fortify their capabilities and integrate novel relationships [7,8]. Yet, they are not without challenges, including the need for timely and accurate entity labeling that may neglect rare diseases or fail to capture the state-of-the-art therapeutic approaches in the fast-changing research environment.

Meanwhile, transformer-based large language models have redefined the possibilities of language processing. These models encompass a deep learning architecture that facilitates the creation of term embeddings, enabling semantically similar terms to converge within a high-dimensional latent space [9]. Moreover, these models not only create term embeddings, but also perceive context, making them proficient at interpreting complex sentences and paragraphs. Examples like GPT-3 [10], ChatGPT [11], T5X [12], and BLOOM [13] have progressively refined their capacity to explore and contextualize vast amounts of existing data. But they, too, aren't immune to criticism, notably concerning data source integrity, topicality, relevance, and potential biases emerging from data curation.

In our case study, we aim to identify and analyze literature corpora of related clinical conditions to search for potential repurposing drugs by employing NLP approaches and establishing a pipeline for real-world validation of the predicted outcomes. We draw inspiration from the work of Tshitoyan *et al.* [14], who pioneered a methodology that mined scientific literature across disciplines, including materials science, physics, and chemistry, encoding the contained knowledge in word embeddings and thereby discovered a new thermoelectric material. We apply a similar methodology to identify drugs for repurposing by utilizing peer-reviewed publications on the meticulously curated repository PubMed (www.pubmed.ncbi.nlm.nih.gov) of the National Institute of Health (NIH), the most prominent biomedical research database. The information therein undergoes rigorous validation by independent experts, establishing its credibility.

With this aim in mind, we developed PubDigest, an NLP-centric software tool that merges the worlds of computational science and biomedicine, leveraging the strengths of both to tackle the challenges of drug repurposing. Chronic thromboembolic pulmonary hypertension (CTEPH), a rare thromboembolic disorder without causative drug treatment [15], presents a fitting case study that demands a deeper exploration for therapeutic interventions. Thus, we aim to identify novel potential drug repurposing candidates for CTEPH that might have escaped traditional research methodologies, and validate the predicted repurposing candidates by *in vitro* tests of anti-thrombotic properties, *in silico* drug toxicity assessment, and real-world clinical data exploration.

## Results

### TF-IDF analysis identifies riociguat as the key medication in CTEPH

CTEPH is a rare, life-threatening disease, characterized by obstructive remodeling of the main pulmonary arteries due to thromboembolism and appearance of organized thrombi [15]. Affecting approximately five individuals per million annually, its primary treatment is surgical embolus removal. In cases where surgery is not an option, or in persistent/recurrent conditions post-surgery, treatment options are limited, underscoring the need for innovative therapeutic strategies. Hence, we chose CTEPH to demonstrate PubDigest's capabilities (Fig 2A and 2B).

To understand the focus of recent CTEPH research, we employed the information gain metric "term frequency-inverse document frequency" (TF-IDF) [16], which gauges the

A)

B)

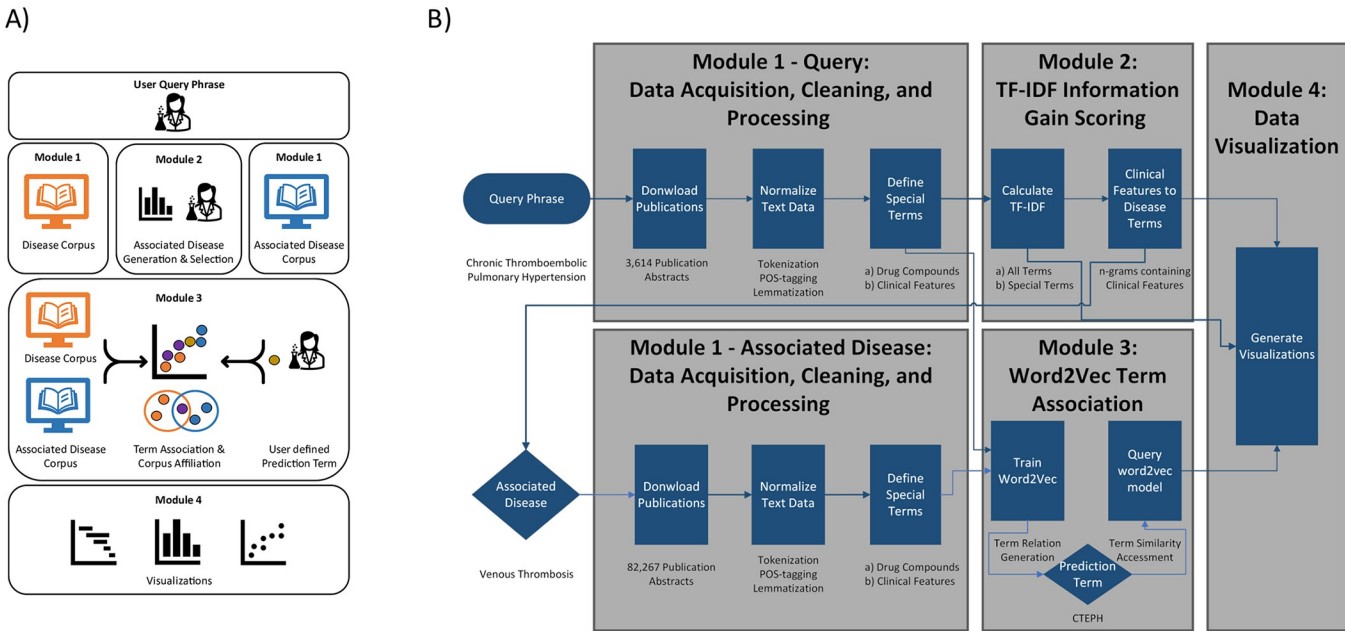

**Fig 2. Program schematics of PubDigest, our NLP-centric tool for drug repurposing.** A) Non-technical program flow chart, providing a general overview. B) Detailed program flow chart, illustrating each module's intricacies. PubDigest's four modules are executed sequentially: data acquisition, cleaning, and processing; TF-IDF information gain scoring; word2vec term association; and data visualization. A user-supplied "Query Phrase", in this case, "chronic thromboembolic pulmonary hypertension", initiates the data acquisition from the PubMed scientific literature meta-data database, engaging the first two modules. The subsequent list of drug compounds and disease terms related to the query are ranked by relevance and frequency. An "Associated Disease" term (here "venous thrombosis") chosen by the user can then be employed for further exploration of literature associations, triggering the repetition of the data acquisition, cleaning, and processing module and generating a second literature corpus. The third module uses the word2vec model to establish term relationships within the two literature corpora. This model can be queried to measure similarities to a user-defined "Prediction Term" (here "CTEPH"), revealing potential term relationships based on context. Lastly, the results are presented visually by the fourth and final module.

relevance of terms within a specific literature corpus. By ranking drug mentions in CTEPH-related literature (3,614 abstracts), corpus-wide TF-IDF highlights key treatments while filtering out common but less relevant terms. Our analysis revealed riociguat as the most noteworthy drug in recent CTEPH studies (Fig 3A), particularly noted for its rising prominence over the past eleven years when compared against the 20 top-scoring candidates (Fig 3B).

Riociguat's dominance in CTEPH literature began after its introduction in 2009 [17,18], as can be seen by the yearly rise in interval-weighted TF-IDF. It effectively replaced earlier off-label vasodilators like sildenafil, bosentan, and iloprost that are ranked second, fourth, and fifth by corpus-wide TF-IDF, respectively [19–21]. Originally developed for treating pulmonary hypertension, riociguat enhances cardiopulmonary hemodynamics by stimulating soluble guanylate cyclase (sGC) activity, drawing parallels with the third-ranked term guanylate (inclusion due to its anti-hypertensive drug prefix "guan-"), which is also the main target of riociguat. Riociguat improves the binding affinity of sGC to nitric oxide (NO), thereby boosting sensitivity to low levels of endogenous NO. This activation of the sGC-NO-cGMP pathway, while effective in addressing CTEPH, also produces wider systemic effects, emphasizing the need for more specific medical therapies.

## TF-IDF tags thrombosis as a principal characteristic of CTEPH

To gain a greater understanding of CTEPH's pathology and clinical representation for drug repurposing opportunities, we used interval-weighted TF-IDF to automatically assess key clinical features associated with CTEPH (Fig 4A). Thrombosis (0.0664), dyspnea (0.0358),

A)

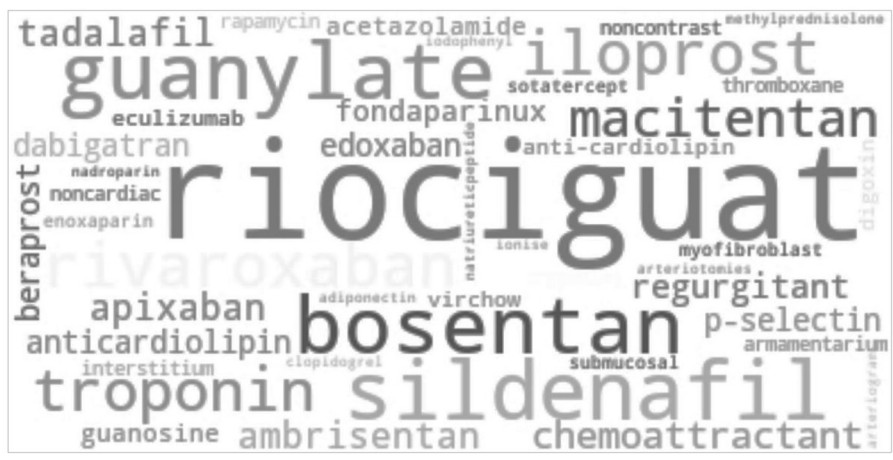

B)

### Interval-weighted TF-IDF drug compound ranking

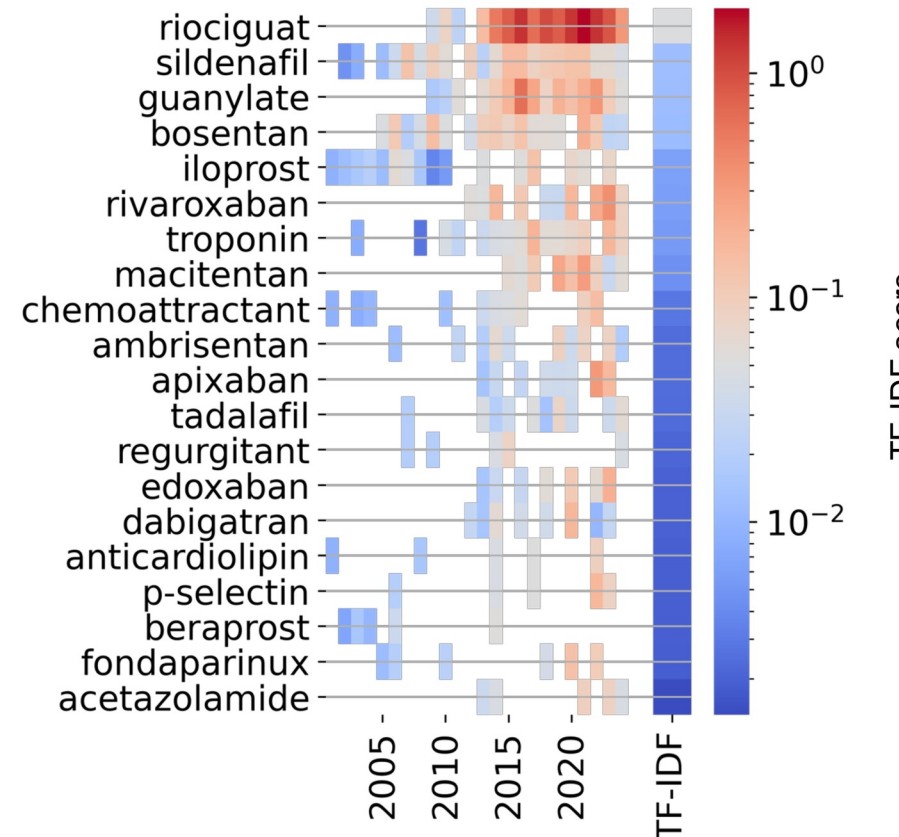

**Fig 3. TF-IDF score highlights riociguat's relevance in CTEPH research.** A) The most notable drugs in CTEPH literature are displayed in this word cloud, with font sizes representing the drug's corpus-wide term frequency on a linear scale, accentuating the prominence of riociguat. Color and opacity do not convey information but enhance readability. B) The interval-weighted TF-IDF reflects the annual interest accrued by a specific compound within CTEPH research. This score is depicted using pseudo-colors, with each drug's overall rank based on its corpus-wide

TF-IDF score. Notably, in years with no publications mentioning a particular compound, corresponding fields in the visualization are left blank, indicating fading research or clinical interest.

thrombolysis (0.0178), fibrosis (0.0158), stenosis (0.0158), and hypoxemia (0.0124) emerged as the top-ranked terms related to CTEPH receiving the highest corpus-wide TF-IDF scores, respectively.

## N-grams link venous thrombosis and CTEPH literature

To identify major or overarching pathomechanisms and related diseases, we compiled a list of disease terms associated with CTEPH's clinical features. All clinical features were used to generate n-grams, a combination of (n-1) words flanking the term of interest (clinical feature) within the CTEPH corpus. We collected n-grams of size two to five words with this approach. These were then ranked by their corpus-wide term frequency within the CTEPH abstracts (Fig 4B) and were predominantly 2-grams. The highest frequencies were observed for "vein thrombosis" and "venous thrombosis" (187 and 112 mentions), followed by "thrombosis pulmonary" (84 mentions), all of which are considered sub-types of venous thromboembolism, underscoring the association between CTEPH and venous thrombosis (VT).

## Word2Vec embedding reveals potential drug candidates for CTEPH

With frequent mentions in the CTEPH literature, our analysis identified VT as closely semantically linked to CTEPH, suggesting overlapping pathomechanisms and/or clinical manifestations. The substantial volume of VT-related studies–almost 23-fold that of CTEPH (3,614 *vs*. 82,267 abstracts)—provides an opportunity for mining potential drug repurposing candidates.

We leveraged the word2vec embedding learning model [22] to map semantic relationships between terms in a multidimensional vector space. This process involved training the model using both the VT and CTEPH corpora. We then depopulated the embedding space vocabulary to only include drug compound terms and the "Prediction Term", CTEPH, which resulted in 839 total drug compounds. Subsequently, we scored these drug compound terms for similarity to the "Prediction Term" using the cosine similarity metrics (Fig 5). Medications emerging from this model were automatically categorized into those already linked to CTEPH (direct term associations, n = 162) and new, unassociated ones (indirect term associations, n = 677) by their presence within the CTEPH corpus.

For our case study, we then curated a list of pro- and anti-thrombotic drugs from drug databases and literature (www.drugs.com, www.mmi.de, www.pubmed.ncbi.nih.gov). The embedding space vocabulary was then queried for this thrombotic drug list to identify drugs in the embedding space with known pro- or anti-thrombotic properties. Of the 162 directly associated drugs, 28 were associated with thrombotic properties (18 anti-thrombotic, 10 pro-thrombotic). Of the 677 indirectly associated drugs, 23 were associated with thrombotic properties (10 anti-thrombotic, 13 pro-thrombotic). The previously calculated cosine similarity scores were then used to rank each compound within its respective group for similarity to the "Prediction Term".

Manual curation was necessary to distinguish positive from negative associations/sentiments (*e.g.*, inducing or inhibiting thrombosis) since the model recognizes contextual space rather than specific drug effects. The aim was to identify drugs with known anti-thrombotic properties not previously mentioned in CTEPH literature but showing a high semantic similarity between CTEPH and VT. Hence, manual curation ensured the relevance and accuracy

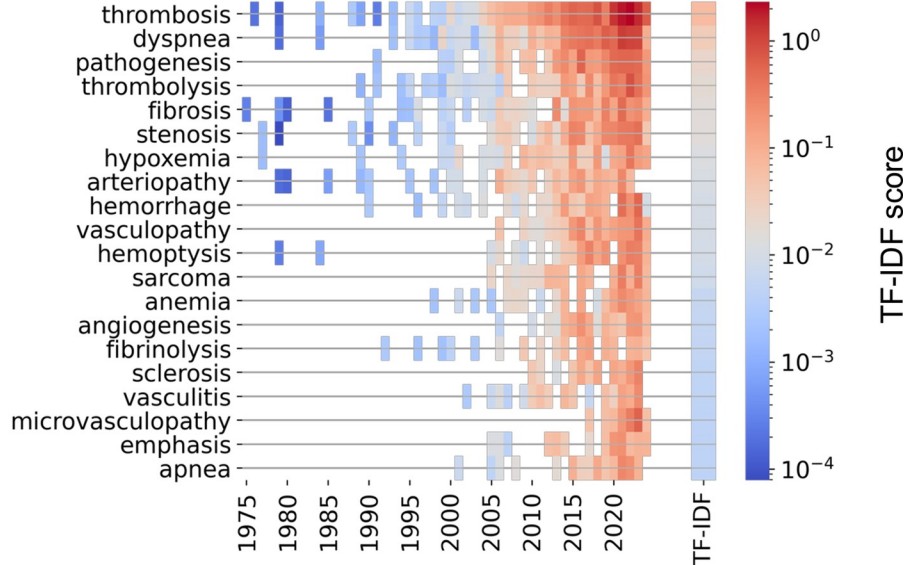

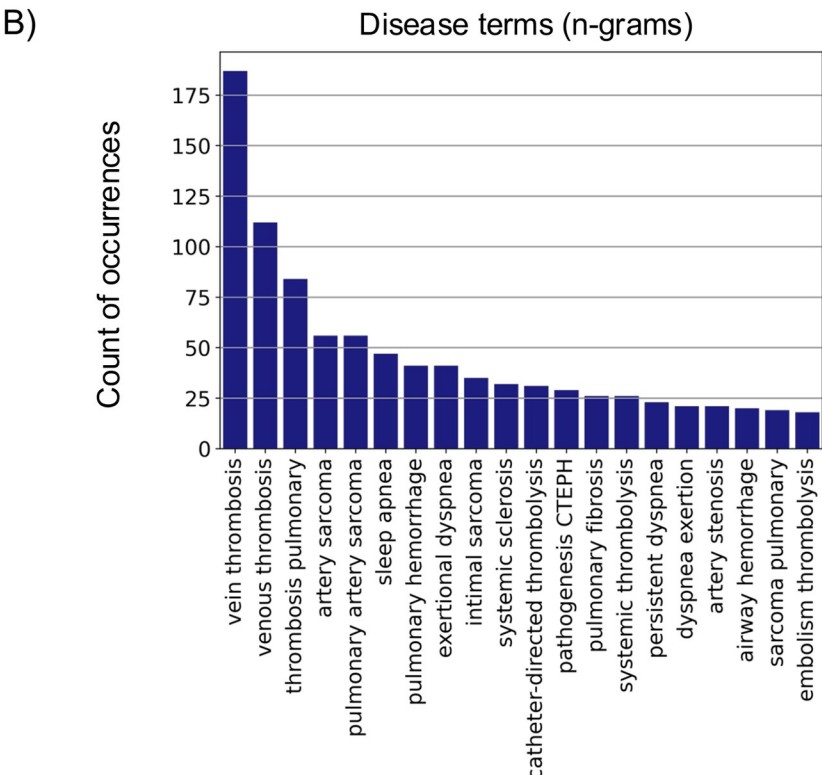

**Fig 4. TF-IDF analysis highlights venous thrombosis as a key disease associated with CTEPH.** A) This section displays the annual interest in specific clinical features of CTEPH, as measured by interval-weighted TF-IDF. Features are ranked based on their corpus-wide TF-IDF score. The TF-IDF values are depicted using pseudo-colors. Absence of data (no mentions) for any year leaves the corresponding field blank. B) The top twenty n-grams including a clinical feature identify the most frequently occurring disease terms in the CTEPH literature. The n-grams have been restricted to n = [2,3,4,5] and a frequency > 5.

Prediction term: CTEPH
Prediction year: 2024

Term occurrence
within corpora

## Direct term association

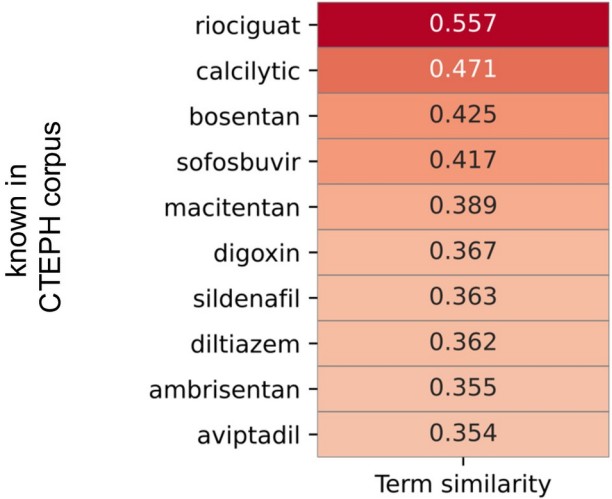
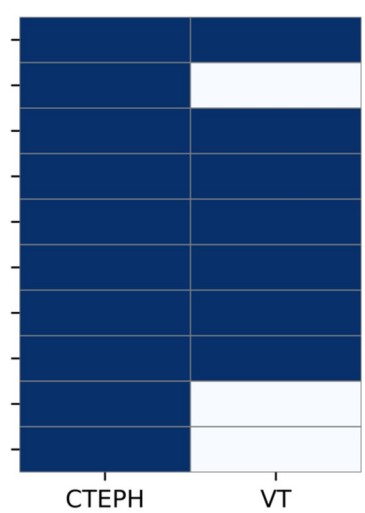

## Indirect term association

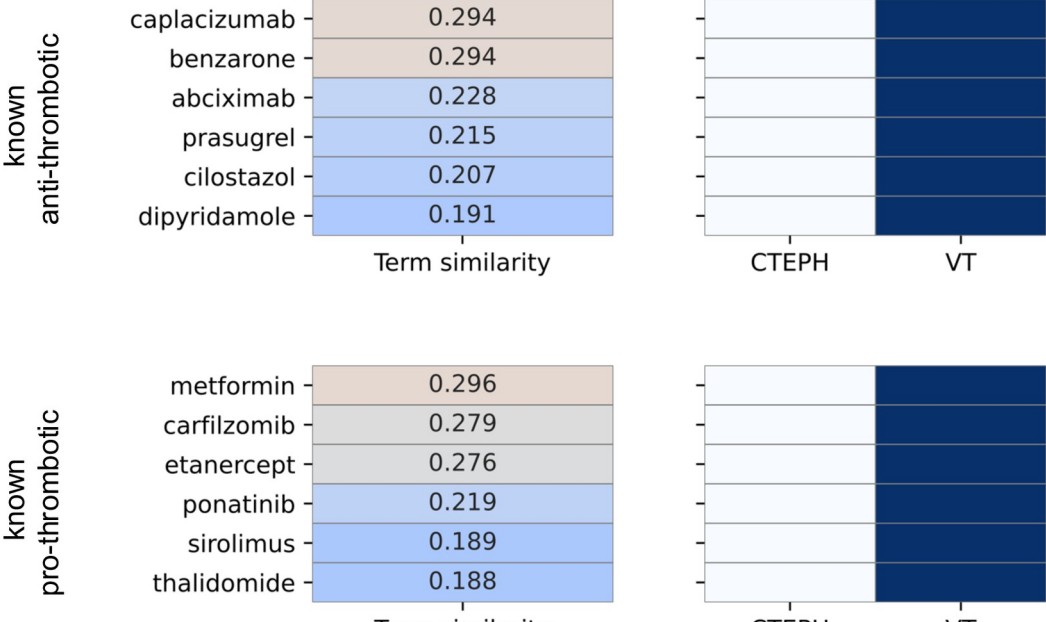

**Fig 5. Word2Vec reveals new links between venous thrombosis medications and CTEPH.** The model ranks drug compounds in relation to the term CTEPH based on cosine similarity and sorts them into direct and indirect term associations (heatmaps). The direct term associations refer to medications found in the CTEPH literature (right panels, dark blue marks their presence in the corpus). The indirectly associated terms were further processed to identify compounds with known pro- or anti-thrombotic properties. Of the 839 drug compounds contained in the embedding space, 162 direct associations have been identified, with the top 10 shown here. Of the 677 indirect term associations, 23 were labeled to have a thrombotic effect (10 anti-thrombotic and 13 pro-thrombotic), with the top 6 candidates shown per group.

of the drug associations identified. No additional manual curation was undertaken on the output.

Caplacizumab and benzarone emerged as top anti-thrombotic medications previously unassociated with CTEPH in existing literature abstracts. Caplacizumab, with a cosine similarity score of 0.294, is approved for treating adult acquired thrombotic thrombocytopenic purpura (aTTP), a rare blood disorder caused by an autoantibody-mediated severe deficit in the von Willebrand factor (vWF) cleaving metalloproteinase ADAMTS13 [23]. The low ADAMTS13 activity results in the formation of ultra-large vWF multimers that bind excessive platelets. Caplacizumab effectively inhibits the interaction between vWF and platelets [24]. However, as we were not granted access to the medication by the manufacturer, caplacizumab was ruled out for our study.

Benzarone, with a matching cosine similarity score of 0.294, also demonstrated a notable semantic connection and was not previously linked to CTEPH in existing literature abstracts, prompting us to explore its viability as an experimental treatment for CTEPH.

## Amiodarone Emerges as a clinically used potential repurposing candidate for CTEPH after *In Silico* toxicological assessment

In the pursuit of a suitable drug repurposing candidate for CTEPH, benzarone stood out from the VT corpus. However, benzarone was removed from clinical usage for its severe hepatotoxic effects [25]. This led us to investigate structurally similar benzarone analogues with a more beneficial toxicity profile, which nicely exemplifies potential challenges for automated drug repurposing pipelines. The transition from benzarone to amiodarone focused on retaining the benzarone structure while prioritizing clinical acceptance and manageable toxicity. This process involved manually identifying analogues with the benzarone structure, considering structural metabolic retention, and prioritizing compounds with well-documented clinical use or significant preclinical data. Benzbromarone, benziodarone, and amiodarone, all belonging to the 1-benzofuran family, were identified by this approach, of which only amiodarone is in clinical use, as a class III antiarrhythmic medication (Fig 6A).

This set of compounds was further refined through an *in silico* toxicology evaluation using our webtool ProTox-II [26], focusing on active toxicity endpoints (Fig 6B). The screening indicated that both benziodarone and benzbromarone share or exceed benzarone's toxicity profile in almost all aspects, prompting their elimination from further analysis. Amiodarone [27], on the other hand, presented with an acceptable toxicity profile. It was predicted to be devoid of active toxicity targets except for high immunotoxicity and a moderate risk of hepatotoxicity. Amiodarone's immunotoxicity, assessed using immune cell cytotoxicity data, shows potential anti-tumor effects at safe doses in cancer cell line studies [28], and the European Medicines Agency (EMA) acknowledges amiodarone's overall benefit-risk profile as positive [29]. Given this favorable assessment, and the need for effective treatments in CTEPH, we selected amiodarone for *in vitro* validation.

## Amiodarone significantly inhibits platelet aggregation *in vitro*

Our investigations into amiodarone, guided by *in silico* predictions, included further assessment of its effects on platelet adhesion and aggregation using our controlled *in vitro* blood flow model [30]. Analysis involved comparing unstimulated and ADP-stimulated whole blood from healthy donors, with or without drug pre-treatment. As a baseline, unstimulated and untreated blood showed 90.9% less platelet adhesion than untreated blood with ADP stimulation.

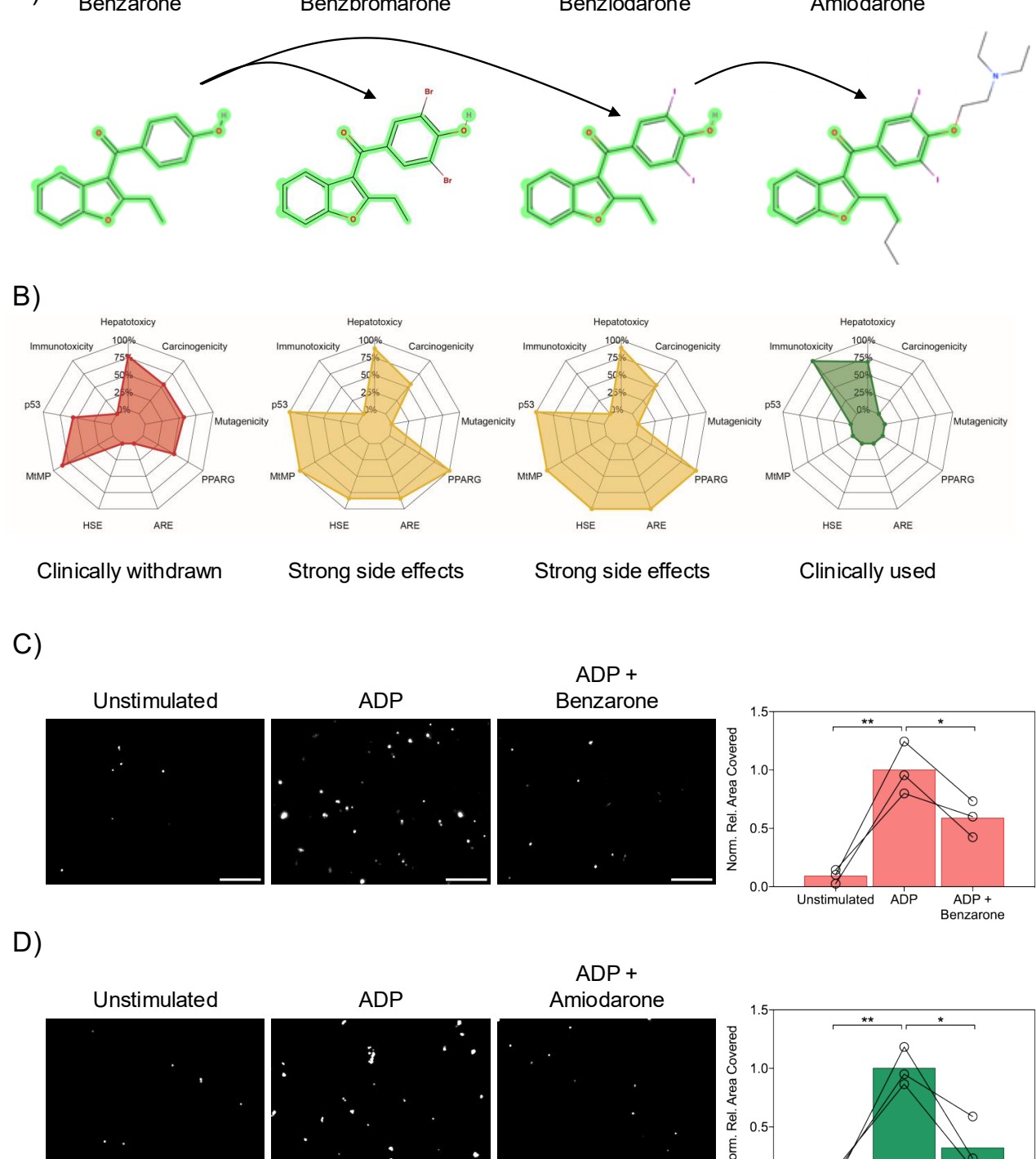

**Fig 6. Amiodarone was identified as a clinically used potential repurposing candidate for CTEPH.** A) Using benzarone as a lead compound, a structural similarity search identified three analogues, including amiodarone. B) ProTox-II predicted toxicity (probability, %) of these compounds, focusing on several toxicity targets and adverse outcome pathways (AOP), such as peroxisome proliferator activated receptor gamma (PPARG), antioxidant responsive element (ARE), heat shock factor response element (HSE), mitochondrial membrane potential (MtMP), and phosphoprotein (tumor suppressor) p53. C) Platelet adhesion to collagen IV surfaces using healthy donor blood at 20 mL/h perfusion for 5 minutes. Blood was pre-

treated with either a vehicle (DMSO) or 5 μM benzarone for 20 minutes, followed by no stimulation or 50 μM ADP stimulation immediately prior to perfusion. D) Repetition of the adhesion assay with 5 μM amiodarone pre-treatment. Utilization of fresh blood from the same donors for C) and D) (n = 3). Platelets visualized by CD42b staining (gray; scale bar = 50 μm). Quantification displayed as percentage of CD42b-positive area relative to total image area, normalized against ADP-stimulated adhesion.

Pre-treatment outcomes varied between the drugs tested. Benzarone (Fig 6C) showed a 41% reduction in ADP-induced platelet adhesion (p = 0.03) to collagen coated surfaces. More significantly, amiodarone (Fig 6D) exhibited a striking 68% decrease in adhesion under similar conditions (p = 0.02), underscoring its potential effectiveness in CTEPH by reducing the likelihood of thromboembolic events.

## Real-world data reveal higher mortality in amiodarone treated CTEPH patients not linked to thrombosis

Finally, we critically evaluated the real-world impact of amiodarone treatment on patients with CTEPH. Utilizing the TriNetX analytics network, which aggregates global clinical data of approximately 40 million patients from 88 healthcare organizations, our study analyzed 453 age- and gender-matched CTEPH patients. All patients analyzed were also suffering from arrhythmias, to identify CTEPH patients potentially treated with amiodarone (Table 1). Amiodarone, primarily prescribed for managing cardiac arrhythmias in these patients, was associated with a significantly higher mortality rate. Statistically, the mortality risk was elevated by 15.9% (p<0.001) in the amiodarone-treated group compared to those not receiving this medication.

Kaplan-Meier survival analysis further revealed a dramatic decline in five-year survival rates for CTEPH patients on amiodarone, dropping from 76.43% to 54.62%, a notable decrease of 21.81%. This increased mortality risk, however, did not appear to correlate with thrombotic complications. There was no significant difference in the occurrence of intracardiac thrombosis, deep vein thrombosis, and pulmonary embolism, or the development of acute myocardial

**Table 1. Comparative outcome analysis utilizing the TriNetX real-world data analytical network for the primary endpoint death.** Analysis was performed after propensity score matching for age and gender at the index event.

| Risk analysis | | | | | |
|---|---|---|---|---|---|
| | Cohort | Patients in cohort | Patients with outcome | Risk | |
| 1 | +arrhythm+cteph +amiodarone | 453 | 139 | 0.307 | |
| 2 | +arrhythm+cteph -amiodarone | 453 | 67 | 0.148 | |
| | | | 95% CI | z | p |
| | **Risk Difference** | 0.159 | (0.105, 0.213) | 5.707 | 0.000 |
| | **Risk Ratio** | 2.075 | (1.598, 2.693) | N/A | N/A |
| | **Odds Ratio** | 2.550 | (1.838, 3.538) | N/A | N/A |
| **Kaplan-Meier survival analysis** | | | | | |
| | Cohort | Patients in cohort | Patients with outcome | Median survival (days) | Survival probability at end of time window (5-years) |
| 1 | +arrhythm+cteph +amiodarone | 453 | 139 | – | 54.62% |
| 2 | +arrhythm+cteph -amiodarone | 453 | 67 | – | 76.43% |
| | | $\chi^2$ | df | p | |
| | **Log-Rank Test** | 29.398 | 1 | 0.000 | |
| | | Hazard Ratio | 95% CI | $\chi^2$ | df | p |
| | **Hazard Ratio and Proportionality** | 2.195 | (1.640, 2.938) | 0.019 | 1 | 0.890 |

infarction and primary pulmonary hypertension between the two cohorts. The full TriNetX report can be found in the Dataverse repository.

One critical adverse effect observed was a significant rise in heart failure incidents among those administered amiodarone, showing a 29.2% increased risk ($p < 0.001$). This study highlights the complexities and potential risks associated with drug repurposing, especially in conditions with multifactorial pathologies like CTEPH.

## Discussion

To find drug repurposing options for CTEPH, our study aimed to elucidate insights into CTEPH medications and its pathophysiological associations with other diseases by combining computational sciences with laboratory testing and clinical evidence. We employed methods including text mining, TF-IDF analysis, word2vec embeddings, toxicology predictions, and real-world data analysis. A pivotal outcome was the confirmation of riociguat as a significantly emphasized drug in recent CTEPH literature, as riociguat is currently the only medication specifically approved for inoperable CTEPH or persisting/reoccurring cases after surgery [31,32]. Our study reinforces its critical role, as evident in recent literature.

Our research also emphasizes a strong semantic connection between venous thrombosis and CTEPH, mirroring findings from a large prospective international CTEPH registry where most patients had histories of acute pulmonary embolism (75%) and deep vein thrombosis (56%), both of which are part of the overarching disease spectrum of venous thromboembolism [33]. Additionally, a longitudinal study from England indicated a 3.5% incidence of CTEPH following VT [34]. These correlations highlight the clinical interconnection between CTEPH and thrombotic events, suggesting that insights from VT management could benefit CTEPH treatment. Furthermore, this emphasizes the capacity of PubDigest to generate relevant semantic connections without the need for prior investigations, specific domain knowledge, or pre-curated database inputs.

This is particularly useful, as previously mentioned state-of-the-art knowledge graph tools are difficult to apply for those without preexisting bioinformatic experience, don't implement the full scope of information necessary for a rare disease like CTEPH, and require pre-curated network files and pre-calculated semantic hierarchy inputs using additional external tools and databases. PubDigest provides a complete question-to-answer tool in a single package, with simple inputs that define three terms of interest: a "Query Phrase", an "Associated Disease" (which is guided by the program), and a "Prediction Term". Furthermore, the usage of the external literature database PubMed ensures comprehensive, up-to-date information is utilized. For a detailed comparison of PubDigest with DREAMwalk [5] and PrimeKG [6], including usability, data exploration, and utility assessments, please refer to the Supplementary Information (S1 Text).

By analyzing scientific abstracts on VT and CTEPH, PubDigest proposed caplacizumab as one of the two top ranked re-purposing candidates alongside benzarone. Caplacizumab is a clinically used nanobody that inhibits the binding of platelets to vWF [23], a central mechanism for primary hemostasis. However, our study was notably constrained by the lack of access to caplacizumab, potentially limiting the applicability of our findings. Additionally, caplacizumab is neither contained in the ProTox-II nor the TriNetX database. Future studies incorporating caplacizumab might provide a broader perspective, particularly in examining its role in managing CTEPH and addressing *in situ* thrombosis through vWF modulation, a concept we and others have earlier suggested [35,36].

Due to this lack of access, we further investigated the benzofuran family, the other top-ranked candidate for drug repurposing in CTEPH. Benzofuran compounds, known for their diverse and strong biological activities including anti-inflammatory and anti-cancer properties

[37], could offer novel mechanisms of action in CTEPH. Specifically, our pipeline suggested amiodarone, an antiarrhythmic medication, for repurposing in CTEPH. Interestingly, amiodarone has been shown to interfere with arterial thrombus formation by modulating surface activity and expression of the tissue factor protein, thereby influencing secondary hemostasis [38]. This function suggests a role beyond its primary antiarrhythmic effects, potentially extending to the modulation of thrombotic processes. However, amiodarone's effects on thrombosis are paradoxical, as differing reports indicate that it can both induce and prevent thrombosis. On the one hand, it can interact with warfarin and enhance its anticoagulant effect, potentially leading to bleeding incidents [39], and can also induce mild thrombocytopenia that occasionally escalates into severe, rapid, and life-threatening drops in platelet counts [40]. On the other hand, it has been associated with cases of extensive thrombosis [41]. Moreover, amiodarone's potential to cause idiopathic pulmonary fibrosis adds another layer of complexity to its use and necessitates careful lung monitoring during treatment [42].

Our *in vitro* studies show that amiodarone significantly reduced platelet adhesion and aggregation, indicating a possible therapeutic effect. However, its ambiguous mechanism at the molecular and cellular levels complicates the prediction of its clinical impacts and requires further mechanistical exploration. This complexity is evident in the real-world data from the TriNetX Analytics Network. This data, covering a large cohort of age- and gender-matched CTEPH patients with arrhythmias, showed a notably higher mortality rate in CTEPH patients with amiodarone-treated arrhythmias, as opposed to those without amiodarone. Notably, the increased mortality risk did not seem to be linked to thrombosis but was accompanied by a rise in heart failure incidents in the amiodarone group. This underscores the complexities and risks involved in repurposing drugs for multifactorial diseases like CTEPH, demonstrating the importance of a comprehensive and careful evaluation of such therapies in a prediction-to-validation pipeline like shown here.

Overall, PubMed-listed abstracts provide a focused, peer-reviewed study summary, but extracting these nuanced insights from abstracts alone has limitations. Including full-text analyses could enhance the depth and accuracy of our findings, despite the challenges of paywall restrictions, varying publisher formats, and retrieval methods. Public databases like PubMed Central (www.ncbi.nlm.nih.gov/pmc), though limited in full-text coverage, might mitigate some of these issues. Furthermore, the incorporation of advanced neural network models such as BERT and its pre-trained version SciBERT [43,44], which understand complex language contexts and specialized scientific vocabularies, as well as improvements to the original word2vec word embedding model [22] that add capacity for sentiment analysis (*e.g.*, positive, and negative associations) [45], could potentially enhance predictive accuracy and model usability.

In conclusion, our research demonstrates the utility of automated analytical tools and language models in systematically examining vast amounts of scientific literature. They enable efficient and objective knowledge exchange across various research fields without the need for specific domain knowledge. While these tools offer groundbreaking possibilities, expert scrutiny is still essential to validate the findings. As we advance, combining such computational capabilities with comprehensive empirical testing and expert analysis will increase accuracy of such tools for finding therapies for complex diseases.

## Material and methods

### Ethics statement

The Ethical Review board of the Charité–Universitätsmedizin Berlin, German approved the collection of blood from healthy volunteer donors, under the ethical approval number EA2-66-20, amendment 3, project 187. Appropriate written informed consent was collected, when applicable.

PubDigest is a Python 3.7 command line application ([www.python.org](www.python.org)). It consists of four modules ([Fig 2A and 2B](Fig 2A and 2B)): data acquisition, cleaning, and processing; term ranking using TF-IDF [16]; term association with the neural network architecture word2vec [22]; and data visualization.

## Data acquisition, cleaning, and processing

We accessed freely available meta-data from the PubMed database. The Entrez API facilitates this data extraction. For a user-defined "Query Phrase", all matching and accessible published meta-data are downloaded. The meta-data includes authors, affiliations, publication date, research field, journal name, title, abstract, keywords, and more, where our focus lies primarily with the analysis of the abstracts. The specific query for this study was "(chronic thromboembolic pulmonary hypertension) AND (english[Language])". Entrez provided 3,624 publication abstracts, of which 3,614 were accessible ([S1A Fig](S1A Fig) and [S1 Table](S1 Table); retrieval date: July 11, 2024).

The text meta-data underwent normalization using NLP tasks included in the *NLTK package* ([www.nltk.org](www.nltk.org)). These tasks encompass tokenization, part-of-speech tagging, and lemmatization, resulting in single terms in their base form. As such, nouns will be in the singular form, while verbs will be in the infinitive, *etc*. Subsequently, we identified special terms by defining two categories: a) Drug compounds curated using the United States Adopted Names (USAN) approved stems, which represent common word stems for which chemical and/or pharmacologic parameters have been established [46]. These guide the naming of new drug compounds belonging to an established series of related agents. To mitigate some of the noise associated with this approach, drug compounds were required to be absent from the standard English dictionary as part of the *PyEnchant 3.2.2 package* ([https://pypi.org/project/pyenchant](https://pypi.org/project/pyenchant)). b) Clinical features or terms matching with a compiled list of suffixes based on a general medical glossary ([S2 Table](S2 Table)).

## TF-IDF information gain scoring

Upon cleaning and processing the data, we continued by scoring the information gain of these terms. The TF-IDF metric [16], facilitated by the *Gensim 4.3.2 package* ([https://pypi.org/project/gensim](https://pypi.org/project/gensim)), allowed us to identify impactful terms in the scientific literature. TF assigns a term-weight proportional to its frequency in a document. IDF estimates the importance of a term in the entire corpus by calculating how frequently a term appears across a corpus, where common terms are given low IDF values, and *vice versa*. Thus, TF-IDF gives a high score to terms that frequently appear in a small number of documents across the entire corpus of literature, reflecting how important a term is in a specific document while considering its use in the entire corpus. The interval-weighted TF-IDF, our iteration of the method ([Formula 1](Formula 1)), calculates the TF-IDF for distinct time intervals, such as specific years or year groups. This method allowed a temporally resolved data representation offering the advantage of promoting terms which were newly added, *e.g.* novel drug compounds. After data acquisition and processing, we applied TF-IDF information gain scoring to each term category. The resulting scores for drug compounds and clinical features were then organized and prepared for visualization. For a clear distinction, the TF-IDF attains the prefix "corpus-wide" to distinguish it from the interval-weighted TF-IDF throughout the text.

$$interval - weighted\ TF - IDF = tf_{interval} \cdot \frac{1}{df_{corpus}} \cdot \frac{D_{corpus}}{D_{interval}} \qquad (Formula 1)$$

tf (term frequency) = number of term occurrences / number of all terms

df (document frequency) = number of documents term appears in / number of all documents in corpus D (number of documents)

## Clinical features to disease terms

In this segment, we generated disease terms using all identified clinical features from the corpus-wide TF-IDF analysis. To do so, n-grams (sequences of 1, 2, 3, or n-adjacent terms) flanking these clinical features were selected. We only considered n-grams with a length of two to five words with a frequency greater than five to minimize noise. Two assumptions have been made: (i) frequently occurring n-grams have a strong association with the original "Query Phrase" and (ii) frequently-occurring n-grams containing clinical features are likely to represent real-world disease terms. With the identification of these "Disease Terms", we then transitioned to studying the relationships between them and the "Query Phrase".

## Word2Vec term association

From the generated list of disease terms, we manually chose the "Associated Disease" ("venous thrombosis") as a targeted approach to this study. Reapplying the first module (data acquisition, cleaning, and processing) for this search term produced a new scientific literature corpus related to VT that consisted of 82,411 abstracts, of which 82,267 were accessible (S1B Fig and S1 Table; retrieval date: July 11, 2024). The third module (term association with the neural network architecture word2vec) then trained the word2vec model using both literature corpora.

Word2vec is an embedding learning/transfer learning technique to generate term associations [22]. This technique utilizes a shallow neural network architecture including an input layer, a single intermediate layer, and an output layer. After the training, the output layer will be removed, revealing a multi-dimensional vector space that is based on the activation functions of the intermediate layer. In this vector space, also called embedding space, terms that are semantically similar are in close spatial proximity, while non-similar terms will be far apart. During the training, words that are used in the same context will be moved closer together. A feature of this model is its capacity to recognize bridging terms: terms that frequently appear in the contexts of two distinct terms that never co-occur. Their presence boosts the perceived semantic relationship between such terms, allowing for novel associations to be uncovered.

The final part of this module involves searching the embedding layer for the "Prediction Term", here "CTEPH", resulting in a list of all terms and their similarities towards the "Prediction Term". This user-defined term should reflect the disease of interest, *e.g.* an often-used abbreviation or a certain characteristic of the disease. For this study, the embedding space vocabulary was reduced to terms in the drug compounds category, which were further labeled by querying for a user-supplied list of drugs that are grouped by one or more factors (here pro- and anti-thrombotic). The model distinguishes between direct term associations (occurrence in CTEPH corpus), which provide insights into currently studied medications, and indirect term associations (no occurrence in CTEPH corpus), yielding potential drug repurposing candidates. The output is ranked by cosine similarity to the "Prediction Term" within the embedding space.

This process has been adapted from the works of Tshitoyan et al. [14], utilizing the same embedding learning method (word2vec), similarity measure (cosine similarity), and definition of direct and indirect term associations. However, we made several significant adjustments. Namely, we identified a specific medical vocabulary (clinical features and disease terms) to build semantic associations and integrated two disease-specific and interrelated medical literature corpora for the term embedding.

## Data visualization

Bar graphs and heat maps were generated using the *matplotlib library* (www.matplotlib.org) and the *seaborn API* (www.seaborn.pydata.org). The word cloud was generated using the

library *wordcloud 1.9.2* ([www.pypi.org/project/wordcloud](www.pypi.org/project/wordcloud)). Having visualized potential drug candidates, we next moved to validate these experimentally.

### Experimental *in vitro* validation

Whole blood of volunteer healthy donors (n = 3) was collected at the Charité–Universitätsmedizin Berlin, Germany with appropriate written informed consent, if applicable, under the ethical approval EA2-66-20, amendment 3, project 187. Donor-matched samples were used to ensure consistency. We followed an established protocol to test agonist-induced platelet aggregation [30]. In short, citrated blood was pretreated with vehicle (DMSO), benzarone, or amiodarone (5 μM) for 20 minutes, re-calcified with Tyrode's buffer containing 2 mM $MgCl_2$ and 2.5 mM $CaCl_2$, left unstimulated or stimulated with ADP (50 μM), and perfused over collagen IV-coated culture surfaces (μ-slides VI 0.4, ibidi) at 20 mL/h for 5 minutes. Subsequently, the flow channels were carefully washed with PBS and fixed with 4% PFA for 20 minutes at room temperature. Platelet adhesion was visualized by staining for CD42b (1:100, Miltenyi Biotec, 130-100-208) and imaged with an EVOS M5000 fluorescence microscope (ThermoFisher Scientific).

### TriNetX real-world database validation

We accessed the TriNetX Analytics Network (retrieval date: June 1, 2023), which provides access to electronic medical records (diagnoses, procedures, medications, laboratory values, genomic information) from approximately 40 million patients from 88 healthcare organizations. We queried this database for patients diagnosed with CTEPH (UMLS:ICD10CM: I27.24), with a primary endpoint of mortality rate. In addition, we comprehensively examined secondary endpoints including a spectrum of cardiovascular complications and thrombotic events. These encompassed intracardiac thrombosis (UMLS:ICD10CM:I51.3), deep vein thrombosis or embolism (UMLS:ICD10CM:I82.4, UMLS:ICD10CM:I82.5), pulmonary embolism (UMLS:ICD10CM:I26, UMLS:ICD10CM:I27.82, UMLS:ICD10CM:I26.9), acute myocardial infarction (UMLS:ICD10CM:I21), primary pulmonary hypertension (UMLS: ICD10CM:I27.0), and heart failure (UMLS:ICD10CM:I50). The analysis timeline was set from one day after the initial diagnosis of the index event (CTEPH and arrhythmia), extending to five years thereafter. Notably, the index events under consideration included occurrences dating back up to 20 years. Our comparative analysis of outcomes differentiated between CTEPH patients with cardiac arrhythmias (UMLS:ICD10CM:I149) who were treated with amiodarone (NLM:RXNORM:703) and those who were not. To ensure robustness in our study, we employed propensity score matching based on age (mean ± SD: 63.8 ± 13.6 years) and gender (50.8% male) at the time of the index event. This matching resulted in the inclusion of 453 patients per cohort. Caplacizumab was not available in the TriNetX database.

### Statistics

The radar plots were created with MATLAB R2022A (MathWorks). Experimental results were confirmed in three healthy donors (biological replicates). Statistics on the experimental results were performed using GraphPad Prism Version 8.3.0. One-way analysis of variance (ANOVA) with Dunnett's multiple comparisons post-hoc test was applied to the normally distributed data. P-values less than or equal to 0.05 were considered significant (* $p \leq 0.05$, ** $p \leq 0.01$, *** $p \leq 0.001$).

## Supporting information

**S1 Fig. Characterization of the Query Phrase and Associated Disease literature corpora.** A) Query Phrase dataset: chronic thromboembolic pulmonary hypertension, B) Associated Disease dataset: venous thrombosis. Overview of number of terms per abstract (y-axis) within the special term categories "Drug Compound" and "Clinical Feature" (hue). Four filters have been added (x-axis): "All Terms" includes all terms within the term category. "Unique Terms" removes all duplications of a term within an abstract. The suffix "No Empty Abstract" removes all abstracts without a term within the term category. Log-scaling of the y-axis enhances visualization of low frequency term occurrences. Retrieval date: July 11, 2024.
(PDF)

**S1 Table. Statistical measures describing the composition of the literature corpora.**
(PDF)

**S2 Table. Compiled list of suffixes used to identify clinical features from the literature corpora.**
(PDF)

**S1 Text. Comparison with Alternative Methods: PrimeKG and DREAMwalk.**
(PDF)

## Author Contributions

**Conceptualization:** Daniel Steiert, Robert Szulcek.

**Data curation:** Daniel Steiert.

**Formal analysis:** Daniel Steiert, Corey Wittig, Priyanka Banerjee, Robert Preissner.

**Investigation:** Corey Wittig.

**Methodology:** Corey Wittig, Robert Szulcek.

**Software:** Daniel Steiert.

**Supervision:** Robert Szulcek.

**Visualization:** Daniel Steiert, Corey Wittig, Robert Szulcek.

**Writing – original draft:** Daniel Steiert, Corey Wittig, Robert Szulcek.

**Writing – review & editing:** Daniel Steiert, Corey Wittig, Priyanka Banerjee, Robert Preissner, Robert Szulcek.

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
