## [Decision Letter · Decision Letter 0]

19 Apr 2024

Dear Prof. Szulcek,

Thank you very much for submitting your manuscript "An Exploration into CTEPH Medications: Combining Natural Language Processing, Embedding Learning, In Vitro Models, and Real-World Evidence for Drug Repurposing" for consideration at PLOS Computational Biology.

As with all papers reviewed by the journal, your manuscript was reviewed by members of the editorial board and by several independent reviewers. In light of the reviews (below this email), we would like to invite the resubmission of a significantly-revised version that takes into account the reviewers' comments.

We cannot make any decision about publication until we have seen the revised manuscript and your response to the reviewers' comments. Your revised manuscript is also likely to be sent to reviewers for further evaluation.

Sincerely,

Fuhai Li

Academic Editor

PLOS Computational Biology

Thomas Leitner

Section Editor

PLOS Computational Biology

Reviewer's Responses to Questions

**Comments to the Authors:**

Reviewer #1: The authors develop a natural language processing system and apply it to identify candidate drugs for use in treating chronic thromboembolic pulmonary hypertension. They select one of the candidate drugs for further assessment, examining its toxicity, in vitro thrombotic and antithrombotic properties, and, via secondary data analysis, risk of death in comparison with matched controls not taking the drug. They thereby illustrate the potential use of natural language processing approaches in identifying and then assessing drugs that may be candidates for repurposing in a specific clinical context.

The authors indicate that they aim to “contribute a novel, efficient pathway to drug discovery”. Indeed, in comparison to other, similar articles in the biomedical informatics literature, this article stands out for its clarity of communication and logical reasoning (it is rare for NLP-centric articles to include follow-up analysis such as in-vitro and drug safety assessment). The background provides a very good summary of relevant literature. The methods are clearly described, and the authors have made their code and data publicly available.

Minor comments:

In the third paragraph of the Results, starting on line 187, the authors indicate that “riociguat enhances cardiopulmonary hemodynamics by stimulating soluble guanylate cyclase (sGC) activity that ranks fourth in our analysis”. The fact that sGC activity ranks fourth in the analysis is difficult to interpret without further context; this should be clarified.

In the second paragraph of the section “Word2Vec Embedding Reveals Potential Drug Candidates for CTEPH”, on p. 9, Word2vec should be cited when it is first mentioned (line 218).

Later in this same paragraph, the authors indicate that “Manual curation was needed, as the model doesn’t distinguish positive from negative associations/sentiments (as in inducing or inhibiting thrombosis).” If there were additional reasons that manual curation was needed these should also be mentioned.

In Figure 1B, in the “Module 1.0” panel, the heading under “Process Bibliometric Data” reads “Influential Authors by Country and City”. However, there is no mention of an analysis related to influential authors in the manuscript. This should either be addressed in the manuscript or removed.

In Figure 2A (the word cloud) the individual words appear to vary with respect not only to size, but also color and opacity. If the variance of these visual properties is meant to convey information, the legend should be updated to reflect this.

Reviewer #2: The paper describes the idea to rely on curated literature from PubMed to draw conclusions, and to combine literature corpora of a related clinical condition to search for potential drugs. The manuscript is an intriguing combination of computational approaches of automatic literature research, toxicity and biological assessment of drug action, as well as studying the drug action using survival and risk analysis in clinical data. The whole pipeline was performed for the case study of a rare disease, CTEPH.

The manuscript is very well written and the big picture is very well understandable. However, it remains extremely shallow both in the presentation of its own results, especially for intermediate results of the literature analysis part, and particularly in relation to state-of-the-art performance of other approaches of literature mining from the literature (that are not performed for the examined case at all). Thus, in contrast to its claims in the introduction, they do not present a new and, especially, demonstrably robustly performing computational approach comparable to or extending the state of the art. Further, many transitions between analysis steps rely heavily on manual curation and thus, its straightforward applicability and usefulness for other cases is unclear. In combination with the very sparse level of description of intermediate results, it is not possible to reproduce the work, nor assess the amount of manual curation required, nor the validity of the manual curation that went into the current study. Please find a more detailed list of required content and clarifications for appropriate assessment below.

Their introduced new measure for relevance, interval-controlled TF-IDF, is used to create nice and insightful figures, but it is does not become clear how this novel aspect was used in their analysis pipeline, and thus the benefit of this new measure is not quantified. Overall, the contribution beyond a case study, arguably very nicely presented with impressively different levels of investigation (from literature to molecular toxicity, wet-lab drug action and clinical outcomes), remains unclear to me. I have the slight impression that the contribution might be oversold as an alternative to, e.g., KG-based approaches, but the benefit in this regard is not even close to be shown.

It may be an excellent option to focus the manuscript on the (really nice) case study with multiple layers of investigations as a result from the start. In any case, supplementing the work by a decisively increased level of description detail is urgently required.

Major points

1) Neither the data repo (claiming to provide intermediate results) nor the software repo are available under the provided links!

2) The work fails to cite newer knowledge graph approaches such as PrimeKG [https://www.nature.com/articles/s41597-023-01960-3] (there are only two citations from 2015 and 2020). Also, what would be the outcome of investigating CTEPH and related drugs with these databases? Is it possible or not possible? This would be an important orthogonal check to determine the benefit of the proposed tool vs. the state-of-the-art.

3) Throughout, manual interventions need to be clearly marked, and their result (outcome of the tool – steps for manual curation and why – result after manual curation) need to be provided to the reader, and potentially the robustness with respect to the choices made need to be assessed. If space becomes an issue, this can well be provided in the Supplementary, but these are important steps of a decision process that need to be conducted when wanting to use this tool for a different case.

Examples for this are

(a) How many and which words did you have of each category of CTEPH abstract (clinical, abbreviations, drug compounds)? How many words were used for the word cloud?

(b) For the n-gram analysis: What were the full results for the analysis output from the pipeline? And what does “delivered the most explicit results” mean in this context? What are detailed intermediate results that are gotten from the pipeline for n-grams and how were they handled? What were the results for n-grams with n>2? Why did you choose the top 10 clinical terms for the n-gram analysis? What would happen using the top 12 or 15 or 30?

(c) Word2vec: What does “narrow the focus to drug terms” mean? Was inference only done on drug terms? Or was also the pre-training affected?

4) Entity recognition was mentioned as a problem of knowledge-based approaches. But also in their approach, entity recognition plays a role. How was ensured that their entities were recognized well? I also see that in the word-cloud, e.g., anti-cardiolipin as well as anticardiolipin (with and without hyphen) appear as separate entities, suggesting a lack of term normalization. Plus, there seems to be “noncardiac” and “chemoattractant” in the list of drug compounds – both suggesting that entity recognition might not have worked well here. Please introduce steps to control for this, or at least comment on why this is not a problem in your case, and provide and describe intermediate output for the reader.

5) TF-IDF: Why this and not some other citation relevance metric, e.g., page rank, in a citation graph? What are the benefits? Is it performing better? Or is it particularly fast? This is a point where control experiments and comparisons with other approaches are missing. And what is the benefit from interval-normalized TF-IDF, except for more resolved figures? The development is not considered here at all for advancing in the pipeline, as far as I can see.

6) Venous thrombosis is quite obviously associated with CETPH. What is the benefit of using venous thrombosis as secondary disease literature corpus instead of, e.g. “(acute) pulmonary embolism” that would have been the most natural choice from clinical experience (as written in the discussion)? Describing this properly would enhance the reasoning on the usefulness of the methods (as in: what do you get by the method that you would not have gotten from the start using a less work intensive than your method). Also, what would happen if using the second best clinical association after venous/vein thrombosis as a second literature corpus?

7) What is the full list of intermediate drug results, both direct and indirect? Without this list and a proper description of what exactly was done for manual curation, the extent of manual curation required to come up with the final list of drugs is not possibly judged. However, this is a crucial step in drug repurposing.

8) The transition from benzarone to the eventually examined drug amiodarone is unclear. I think there could be many other derivates with similar molecular structures. Why were these not chosen? This is important for being able to transfer this analysis to other cases.

9) It remains unclear in how far and for which steps the cited approach of Tshitoyan et al. is followed. The relations between the contributions here to the existing work need to be made clearer.

Minor:

10) The abbr. TF-IDF is not introduced in the main text, only in the abstract, and it is not introduced in the results at all (only in the methods that come last in the document)

11) Fig. 1 is cited very last in the main text

12) In the pipeline (Fig. 1), there is a step on “finding influential authors and cities” - What is the benefit of influential authors in the field? Is the literature list extended by this? This step is not described nor mentioned at all in the manuscript.

13) It could be interesting for a reader which clinical studies/data resources could be used if they want to apply the approach to other cases.

14) What about examining derivatives or similar drugs for the first-scoring drug, caplacizumab? Why were these not scrutinized? Also without biological experiments, it would be very instructive for the reader to investigate the clinical impact of this drug according to the large clinical cohorts.

**Have the authors made all data and (if applicable) computational code underlying the findings in their manuscript fully available?**

Reviewer #1: Yes

Reviewer #2: **No: **The provided links to the code and data are not functional. Also, intermediate data and description of their processing needs to be provided for ensuring reproducibility (see details in my response to the authors).

PLOS authors have the option to publish the peer review history of their article (what does this mean?). If published, this will include your full peer review and any attached files.

Reviewer #1: **Yes: **Michael E. Bales

Reviewer #2: No
---

## [Decision Letter · Decision Letter 1]

9 Jul 2024

Dear Prof. Szulcek,

Thank you very much for submitting your manuscript "An Exploration into CTEPH Medications: Combining Natural Language Processing, Embedding Learning, In Vitro Models, and Real-World Evidence for Drug Repurposing" for consideration at PLOS Computational Biology. As with all papers reviewed by the journal, your manuscript was reviewed by members of the editorial board and by several independent reviewers. The reviewers appreciated the attention to an important topic. Based on the reviews, we are likely to accept this manuscript for publication, providing that you modify the manuscript according to the review recommendations.

Sincerely,

Fuhai Li

Academic Editor

PLOS Computational Biology

Thomas Leitner

Section Editor

PLOS Computational Biology

Reviewer's Responses to Questions

**Comments to the Authors:**

Reviewer #1: The authors have adequately addressed my feedback.

Reviewer #2: Review is uploaded as attachment.

**Have the authors made all data and (if applicable) computational code underlying the findings in their manuscript fully available?**

Reviewer #1: Yes

Reviewer #2: **No: **As detailed in the comments, sufficient dataset descriptions and the wordcloud basic data is missing, but otherwise widely available (minor additions required).

PLOS authors have the option to publish the peer review history of their article (what does this mean?). If published, this will include your full peer review and any attached files.

Reviewer #1: **Yes: **Michael E. Bales

Reviewer #2: No

Figure Files:

Data Requirements:

Reproducibility:

References:

---

## [Editor Report · Decision Letter 2]

14 Aug 2024

Dear Prof. Szulcek,

We are pleased to inform you that your manuscript 'An Exploration into CTEPH Medications: Combining Natural Language Processing, Embedding Learning, In Vitro Models, and Real-World Evidence for Drug Repurposing' has been provisionally accepted for publication in PLOS Computational Biology.

Best regards,

Fuhai Li

Academic Editor

PLOS Computational Biology

Thomas Leitner

Section Editor

PLOS Computational Biology

---

## [Editor Report · Acceptance letter]

27 Aug 2024

PCOMPBIOL-D-23-01945R2 

An Exploration into CTEPH Medications: Combining Natural Language Processing, Embedding Learning, In Vitro Models, and Real-World Evidence for Drug Repurposing

Dear Dr Szulcek,

I am pleased to inform you that your manuscript has been formally accepted for publication in PLOS Computational Biology. Your manuscript is now with our production department and you will be notified of the publication date in due course.

With kind regards,

Zsofia Freund
